# Carbon emission quantification analysis of excavation engineering under road transport conditions

**Bai Chongxi** [1] *, **Zhiheng Zhang** [2]

1 College of Architecture and Energy Engineering, Wenzhou University of Technology, Wenzhou, Zhejiang, China, 2 College of Engineering, Northeast Agricultural University, Harbin, Heilongjiang, China

* bcxzcx@163.com

**Data Availability Statement:** All relevant data are within the manuscript and its Supporting information files.

**Funding:** This work was supported by the [ National Natural Science Foundation of China ]

## Abstract

Current research on building carbon emissions primarily focuses on various carbon emission assessment models and the use of life cycle analysis to evaluate overall building carbon emissions, with limited attention given to excavation engineering. Based on the life cycle method and process analysis, this study analyzes carbon emissions in excavation engineering by optimizing the evaluation model for fuel consumption standards of freight vehicles during the transportation phase in China. To account for the difference between actual and rated fuel consumption of transport vehicles, factors such as road conditions, traffic congestion, and temperature are introduced to adjust the carbon emission calculation model for the transportation phase. This approach reasonably incorporates the impact of fuel consumption during vehicle idling on carbon emission calculations. Using the 02B excavation of the Beijing Sub-Center Station transportation hub as a case study to validate the proposed method, the analysis reveals that the primary source of carbon emissions in excavation engineering is earthwork transportation, accounting for 40.50% of total emissions. Among these, earthwork transportation contributes 95.28% of emissions within the transportation phase. Due to adjustments in the carbon emission calculation model for the transportation phase, carbon emissions increased by 1,226.79 tons, accounting for 9.2% of the total. The revised model provides a theoretical basis for accurately assessing carbon emissions in excavation engineering.

## 1 Introduction

With the acceleration of industrialization, the development of society heavily depends on the combustion of fossil fuels, which leads to the emission of large amounts of greenhouse gases, resulting in the intensification of the global warming problem [1]. In the Sixth IPCC Assessment Report, it is noted that compared to the average temperature between 1850 and 1900, the surface temperature has increased by approximately 1.1˚C. In addition, according to the Climate Sensitivity Index (CSI), it is predicted that the global average temperature will increase by approximately 3˚C without any change in the existing global development pattern [2]. In

under Grant [No. 51608101]. The funders had no role in study design, data collection and analysis, decision to publish, or preparation of the manuscript.

**Competing interests:** The authors have declared that no competing interests exist.

response, since 1990, countries worldwide have implemented energy conservation and emission reduction actions on a large scale [3], aiming to control global warming below the 2°C warning line [4].

Energy consumption and carbon emissions originating from the construction industry account for 40% and 36%, respectively, of the total emissions [5]. China, as the world's largest energy consumer and carbon emitter, produces approximately 10 billion tons of carbon emissions annually [6], accounting for approximately 30% of the global emissions [7], of which the carbon emissions stemming from the construction industry account for approximately 35%–50% of China's total carbon emissions [8]. Thus, energy-saving and emission reduction efforts in the construction industry could help China quickly realize the dual-carbon goal. Moon et al. [9] found that the application of the process analysis method at the design stage only produced an error of 8% in calculating building carbon emissions. Keoleian et al. [10] evaluated the life cycle energy consumption and greenhouse gas emissions of residential buildings and found that the energy consumption of the operation and use phase exceeded 91% overall, which is the main source phase of carbon emissions. Yujie Cang [11] proposed a carbon emission calculation method based on the structure of the building as the basic unit to calculate the amount of building materials used at the materialization stage and the associated carbon emissions.

With global urbanization on the rise, limited urban surface space and increasing traffic congestion have become critical obstacles to further urban development. Metro systems, with their high-speed operations and large passenger capacity, are emerging as the optimal solution for many countries to address these challenges, becoming the preferred mode of modern urban transportation [12, 13]. As of early 2022, a total of 36854.2 kilometers of urban rail transit systems have been built across 541 cities in 76 countries worldwide [14]. As of early 2024, mainland China has established urban rail transit systems in 59 cities, with 338 operational lines spanning a total length of 11,224.54 kilometers. Among these, metro lines constitute a significant 76.11%, covering 8,543.11 kilometers in operational length [15]. Given the vast scale of underground space development, accurately assessing carbon emissions in foundation pit projects is essential due to the complex underground environment, constraints on aboveground conditions in urban areas, and the considerable construction challenges. Subway construction, in particular, often requires extensive excavation, which can lead to significant environmental impacts.

Foundation pit construction is the first step in building and rail transit projects; however, there has been limited research on the characteristics of carbon emissions specific to foundation pit engineering. Furthermore, existing carbon emission assessment models have not effectively accounted for the relevant influencing factors. This study, based on lifecycle assessment and process analysis methods, reasonably considers the impact of vehicle unladen conditions on the carbon emission calculation model for the transportation phase. It aims to accurately evaluate the carbon emissions associated with the construction of foundation pits, conduct a comparative analysis of the carbon emissions across various stages of the lifecycle of foundation pit projects, summarize the carbon emission characteristics of such projects, and provide a theoretical basis for energy-saving and emission-reduction measures in foundation pit construction.

## 2 Research method

The research process of this study can be divided into four stages. First, through a literature review and data survey, it highlights the proportion of carbon emissions contributed by the construction industry globally, and emphasizes the growing necessity of analyzing the carbon

emissions of foundation pit engineering as urbanization continues. With the saturation of above-ground construction and the rising trend of rail transit development, this shift further underscores the importance of such an analysis. Next, a life cycle assessment (LCA) is used to analyze the carbon emissions of foundation pit engineering. The calculation model adopts the process analysis method, which is further optimized by incorporating the fuel consumption standards for freight trucks in China. Specifically, the transportation stage model within the process analysis is adjusted to account for the discrepancy between actual and rated fuel consumption. Factors such as road conditions, traffic congestion, and temperature are introduced to refine the model. Additionally, the impact of empty-load fuel consumption on carbon emissions is reasonably considered. The optimized model can accurately assess the carbon emissions during the transportation phase. Finally, the impact of the model adjustments on carbon emissions is demonstrated through the case of the foundation pit for the 02B station at the Beijing Sub-Center Transportation Hub. The carbon emission characteristics of foundation pit engineering are then analyzed. The study concludes by identifying the existing issues and limitations, and outlining the potential directions for future research and development.

## 3 Theoretical analysis of carbon emissions resulting from pit engineering

Carbon emission analysis research mainly uses life cycle assessment (LCA). The quantitative models employed include the direct measurement method, process analysis method, input-output method, and hybrid method [16]. Among these, the process analysis method is the most widely used due to its accuracy and convenience in calculations. Life cycle assessment is a theoretical method used to evaluate the environmental impacts and resource consumption throughout the entire life cycle of a product or production process. It involves goal definition, scope boundary setting, inventory analysis, impact assessment, and interpretation of results to provide a comprehensive analysis of the overall environmental impacts of a building. This approach helps to avoid focusing on only one phase, ensuring a holistic assessment of the environmental impacts during the construction process.

### 3.1 Pit engineering life cycle

The life cycle assessment technique is a theoretical method for assessing the environmental loads and resource consumption levels generated throughout the entire process of a product or production process [17], which suggests a type of cradle-to-grave environmental impact assessment approach. In the construction field, the life cycle of a building can be divided into four phases: production and transportation of building materials, construction, operation and use of the building, and demolition of the building. In the case of pit construction, pit construction does not involve the operation and use phase, while pit demolition is considered in the overall building demolition process. For this reason, the life cycle of a pit project can be divided into three phases: production of building materials, transportation of building materials and pit construction.

### 3.2 Process analysis method

In the process analysis method, the production process is first divided by the work procedure, the product of the corresponding activity data and the carbon emission coefficient is then adopted as the carbon emissions of each subsegment, and the carbon emissions of all subsegments are finally summed to obtain the total carbon emissions of the production process. The

calculation model can be expressed as follows (Eq 1) [18]:

$$E = \sum_{i=1}^{n} (\varepsilon_i \times Q_i) \tag{1}$$

where $E$ denotes the total carbon emissions of a certain production process (t); $\varepsilon_i$ is the carbon emission coefficient of the ith production subsegment; $Q_i$ denotes the activity data of the ith production subsegment; n is the number of production subsegment divisions; and i denotes a certain production subsegment, for i = 1, 2, 3,. . ., n.

## 3.3 Computation Model

### 3.3.1 Carbon emission calculation model for the building material production stage.

In the building life cycle, the determination of the carbon emissions at the building material production stage should consider the carbon offset generated by the dismantling and recycling of materials. Referring to overseas engineering applications, steel sheet piles and steel supports are generally used more than 30 times before they are recycled, and both the carbon emissions and energy consumption of each repair step account for approximately 5% of the total carbon emissions and energy consumption, respectively. Carbon offsets that do account for the reuse of components can cause inflation of the total carbon emissions. Materials other than metals, such as concrete, glass, and wood, can be dismantled and recycled and processed to form new products, and the recycling rates for the main materials are listed in Table 1. Research shows that metal waste has a higher recycling value compared to other types of waste, such as bricks and stones. For example, aluminum products, which make up 0.66% of the total weight, can contribute to a 45% reduction in carbon emissions and provide greater environmental benefits [19]. The recycling rate of aluminum typically exceeds 80%, and the energy consumption for recycling aluminum is only 5% of that required to produce the same weight of primary aluminum. While the extraction and processing of aluminum release significant greenhouse gases, recycling aluminum can greatly reduce emissions. From an economic perspective, due to its high recycling rate and low energy consumption, the market potential for aluminum recycling is promising. Concrete, as a major construction waste, typically has a recycling rate of around 50%, but its recycling process is relatively complex. The production of concrete is associated with high greenhouse gas emissions, and cement production, in particular, contributes significantly to emissions. Although recycled concrete can reduce the demand for new concrete, its mechanical properties are generally slightly inferior to those of virgin concrete.

Considering the material recovery rate, for which correction factor $a_i$ is introduced, the carbon emission calculation model for the production stage of building materials can be expressed as follows (Eq 2) [20]:

$$E_m = \sum_{i=1}^{n} [\varepsilon_i \times Q_i (1 - \alpha_i)] \tag{2}$$

where $E_m$ denotes the total carbon emissions (t) at the material production stage and $a_i$ is the recovery rate of the ith material. The data are provided in Table 1.

Table 1. Main material recovery rates [20].

| makings | concrete | steel reinforcing bar | aluminum | fiberglass | lumber | steel | other metals |
|---|---|---|---|---|---|---|---|
| recovery rate | 50% | 30% | 80% | 50% | 50% | 90% | 90% |

**3.3.2 Model for calculating the carbon emissions at the transportation stage.** The carbon emissions during the transportation phase mainly result from vehicle fuel consumption, and more than 95% of vehicles rely on fossil fuels [21]. Building materials are mainly transported along road, railroad and water transport networks, and we study the carbon emissions of foundation pit projects under road transportation conditions in this paper. The traditional fuel consumption calculation method for truck operation is based on the rated fuel consumption of the vehicle, but under the influence of external factors [22], the actual consumption of the vehicle is often higher than the rated consumption, which leads to notable underestimation of the actual carbon emissions in the traditional calculation method. Therefore, according to the guidelines in Cargo Vehicle Operation Fuel Consumption (GB/T 4352–2022) [23] for optimizing the calculation of carbon emissions at the transportation stage, the actual fuel consumption of cargo vehicle operation can be calculated with Eqs 3 and 4:

$$Q_r = \left( Q_k \times \frac{S_i}{100} + Q_b \times \frac{\Delta G \times S_i}{100} \right) \times K_r \times K_t \times K_v \times K_x \tag{3}$$

$$Q_b = \frac{Q_m - Q_k}{M_m - M_k} \tag{4}$$

where $Q_r$—Actual fuel consumption (L) for a certain mode of cargo vehicle operation;

$Q_b$—Fuel consumption of cargo vehicles per unit load mass change (L/100 km);

$Q_m$—Full load fuel consumption of cargo vehicles (L/100 km);

$Q_k$—Vehicle no-load base fuel consumption (L/100 km);

$M_m$—Maximum gross mass of the cargo vehicle (t);

$M_k$—Full mass of the cargo vehicle (t);

$S_i$—Mileage (km) of the cargo vehicle for a certain mode of operation;

$\Delta G$—Mass of the loaded vehicle (t) for a certain mode of transportation;

Kr—Road correction factor (refer to Table 2);

$K_t$—Temperature correction factor (refer to Table 3);

$K_v$—Congestion correction factor (refer to Table 4); and

$K_x$—Correction factor for other influencing factors, mainly including the car walking-in period, driving practice period, local rainy period, loading of hazardous materials, overturning, and icy and snowy roads, which are generally specified by the car-using organization.

**Table 2. Correction factors for the various road categories [23].**

| Type of road | Category 1 roads | Category 2 roads | Category 3 roads | Category 4 roads | Category 4 roads | Category 6 roads |
|---|---|---|---|---|---|---|
| Kr | 1.00 | 1.10 | 1.25 | 1.35 | 1.45 | 1.70 |

**Table 3. Temperature correction factors [23].**

| Monthly average temperature (°C) | t≤-25 | -25<t≤-15 | -15<t≤-5 | -5<t≤5 | 5<t≤28 | t>28 |
|---|---|---|---|---|---|---|
| Kt | 1.13 | 1.09 | 1.06 | 1.03 | 1.00 | 1.02 |

**Table 4. Congestion correction factors [23].**

| Average travel speed (km/h) | V≤20 | 20<V≤30 | 30<V≤40 | 40<V≤50 | V>50 |
|---|---|---|---|---|---|
| $K_v$ | 1.30 | 1.15 | 1.00 | 0.90 | 0.80 |

According to Eqs 3 and 4, the actual fuel consumption of a vehicle is mainly affected by the type of road, weather and temperature conditions, degree of traffic congestion (defined by the average speed), and load. Regarding the influence of the load, first, we must determine whether the maximum total mass of the vehicle reaches the critical condition of 3500 kg. Second, we must assess the data provided by automobile manufacturers, Road Transportation Vehicle Compliance Vehicle Model List or Fuel Consumption Labeling for Light-Duty Vehicles of the Ministry of Transportation and Communications. Finally, we must calculate the change in the fuel consumption caused by the load. In this study, we show that the no-load fuel consumption of the vehicle is 0.67 times the full-load fuel consumption [24]. Based on this finding, Eqs 3 and 4 are corrected, and the calculation models are expressed in Eq 5:

$$Q_t = \sum_{i=1}^{n} m_i \times 1.67 d_i \times a_i \times K_r \times K_t \times K_v \times K_x \times T_c \tag{5}$$

where $Q_t$ denotes the total carbon emission at the transportation stage (t); $m_i$ is the transportation volume of the ith material (t); $d_i$ is the transportation distance of the ith material (km); $a_i$ is the carbon emission coefficient of the ith energy source; and $T_c$ is the energy consumption per unit of the material transportation distance (MJ-(t-km)$^{-1}$).

**3.3.3 Carbon emission calculation model for the construction phase.** Carbon emissions during the construction phase primarily come from the energy consumption of construction machinery, with gasoline, diesel, and electricity being the main sources. The emission factors are selected based on the emission factor database, and the specific values will be provided in the next section. The calculation model is shown in Eq 6 [25]. It is important to note that the actual energy consumption of construction machinery is also influenced by factors such as workload, operating methods, maintenance conditions, construction environment, and construction techniques. By optimizing these factors, fuel consumption can be effectively reduced, construction efficiency can be improved, and environmental impact can be minimized. Since these factors are influenced by the specific conditions of the construction site, the influencing coefficients are determined by the construction company.

$$E_c = \sum_{i=1}^{n} S_i \times P_i \times EF_i \times K_s \tag{6}$$

where $E_c$ denotes the total carbon emissions (t) during the construction phase; n denotes the types of machinery needed for construction; $S_i$ is the number of shifts needed for the ith type of construction machinery; $P_i$ is the amount of energy consumed per shift by the ith type of construction machinery; $EF_i$ is the energy carbon emission factor; and $K_S$ is correction factors for other influencing factors on energy consumption of construction machinery.

## 4 Engineering examples

### 4.1 Project description

The Beijing Urban Vice Center Station Traffic Hub Project is adopted as an example of analyzing the carbon emissions of a foundation pit project under road transportation conditions. The total construction area of this project is 192000 m$^2$, and the underground construction area is 188644 m$^2$. In the project, the underground part of the 02B foundation pit entails a frame structure, with 2–3 stories, an east–west length of 258.45 m, a north–south width of 141 m, and a total area of 35528 m.$^2$. The foundation pit involves the adoption of the open excavation method, the upper depth is 8 m, and the support method entails the combination of slope

release and soil nail walls. The lower depth ranges from 14.1~17.5 m, and the support method involves the combination of diaphragm walls and steel anchor cables [26].

## 4.2 Calculation of carbon emissions

**4.2.1 Parameter values.** The three main types of building materials transported for excavation engineering purposes include earth and stone, concrete, and other building materials such as steel.

*(1) Transportation distance.* According to the Construction Carbon Emission Calculation Standard (GB/T 51366–2019) [27], the default transportation distance of concrete is 40 km, the default transportation distance of other construction materials is 500 km, and the transportation distance of earth and rocks to the dumping ground should be less than 25 km by accounting for the budget quota of each province and city and the economy. In this paper, this distance is set to 20 km.

*(2) Road correction factor.* As the considered pit project is located in the city, the earth and concrete transportation distances are smaller than that of other construction materials, mainly entailing urban road transportation, urban roads are category 2 roads, and the Kr value is 1.1. Steel and other building materials are mainly transported by roads, and the Kr value is 1.0 for category 1 roads.

*(3) Temperature correction factor.* The temperature interval of the China Statistical Yearbook 2022 is assessed according to the project location area, the temperature correction factor $K_t$ is determined according to Table 3, and the vehicle temperature correction factor $K_t$ in Beijing is set to 1.0.

*(4) Other influencing factors.* Assuming no other influences on transportation, $K_x$ is set to 1.0.

*(5) Congestion correction factor.* According to public data of the Beijing Municipal Bureau of Urban Management and Comprehensive Administration and Law Enforcement, the speed limit of dump trucks in urban areas is 30 km/h, so the congestion correction coefficient $K_v$ for earth transportation is set to 1.15. The speed limit of concrete mixer trucks in urban areas is 40 km/h, so the congestion correction coefficient $K_v$ for concrete transportation is calculated as 1.0. Other building materials are mainly transported by highways, the traveling speed is higher than 50 km/h, and the congestion correction coefficient $K_v$ is set to 0.8.

(1) (6) Materials, energy and labor

Data of the materials, energy and labor consumed per unit of the project volume are obtained from the Beijing Construction Project Pricing Basis–Budget Consumption Standard [28] and the National Unified Machinery Shift Cost Quota [29].

**4.2.2 Calculation results.** The 02B pit bill of quantities and the corresponding carbon emissions are provided in Table 5.

Carbon emission factor data are obtained from the National Greenhouse Gas Inventory Guidelines (IPCC 2006) [30] and the Standard for Calculating Carbon Emissions from Buildings, as shown in Table 6.

## 4.3 Analysis of the results

The carbon emissions during the transportation phase of building materials in pit engineering are slightly higher than those during the production phase of building materials and much higher than those during the construction phase, accounting for 42.51% of the total carbon emissions, as shown in Fig 1. This occurs because the earth excavation amount in pit construction is large and the total amount of transportation is large, which leads to higher carbon

**Table 5. Carbon emission calculation results for the 02B pit.**

| Subengineering | Construction project | Job description | Project volume | Carbon emissions (t) | | |
|---|---|---|---|---|---|---|
| | | | | Building material production phase | Building material transportation phase | Construction phase |
| Earthwork | Site formation | Level the land | 4102.76m³ | 0 | 0 | 7.48 |
| | Excavators | Open cut excavation | 666861m³ | 0 | 5823.88 | 990.89 |
| | Landfill | Landfill | 7901m³ | 55.42 | 0 | 6.00 |
| Concrete and reinforced concrete | Reinforcing steel for cast-in-place elements | Reinforcing steel fabrication | 57t | 14.00 | 2.97 | 0.34 |
| | | Reinforcing steel installation | 57t | 5.26 | 0 | 2.90 |
| | Reinforcement cage (diaphragm wall) | Diaphragm wall Diaphragm wall reinforcement | 1603.531t | 2849.42 | 83.55 | 233.36 |
| Foundation treatment and slope support | High-pressure gusher pile | Rotary drilling | 2416.23m³ | 0.74 | 0 | 126.48 |
| | | Rotary bored piles filled with concrete | 2416.23m³ | 413.95 | 33.24 | 5.94 |
| | | Postcompaction slurry at the bottom (side) of the column | 24.869t | 8.90 | 1.30 | 0.41 |
| | Diaphragm wall | Diaphragm wall guide wall excavation | 361.83056m³ | 0 | 3.16 | 0.79 |
| | | Diaphragm wall guide wall concrete | 361.83m³ | 54.17 | 4.98 | 0.20 |
| | | Diaphragm wall guide wall reinforcement | 52.96t | 93.08 | 2.76 | 4.52 |
| | | Diaphragm wall Diaphragm wall trenching | 8512.32m³ | 0.46 | 74.34 | 610.74 |
| | | Concrete for the diaphragm walls | 8512.32m³ | 1458.34 | 117.10 | 38.90 |
| | Anchor | Prestressing anchor drilling | 17188m | 61.69 | 0 | 51.03 |
| | | Prestressing anchor grouting | 17188m | 532.79 | 10.88 | 22.48 |
| | | Prestressing steel strand | 17188m | 25.96 | 5.70 | 72.38 |
| | Soil nail support | Soil nail fabrication and installation | 11.314464t | 19.32 | 0.59 | 0.57 |
| | | Soil nail drilling and grouting | 3796.8m | 46.64 | 0 | 13.60 |
| | Shotcrete | Shotcrete | 1486.13m³ | 378.98 | 20.44 | 83.67 |
| Project measures | Temporary steel support | Steel pipe fabrication for steel supports | 171.29t | 67.91 | 8.92 | 9.32 |
| | Temporary steel support removal | Installation and removal of steel supports | 171.29t | 1.77 | 0 | 7.27 |
| Total carbon emissions (t) | | | | 6088.8 | 6193.81 | 2289.27 |

emissions at the transportation stage than at the other stages. Among them, the carbon emissions of earth transportation account for 40.50% of the total carbon emission, which is 95.28% of those at the transportation stage, as shown in Fig 2.

The change in the total carbon emissions during the transportation phase due to the modification of the carbon emission calculation model is shown in Fig 3, which indicates that fuel consumption underestimation will result in an error of 1,226.79 tons in the calculated carbon emissions, representing an error of up to 9.2%. According to the carbon trading price of the Shanghai Environmental Energy Exchange in July 2023, the carbon emission quota is 61 yuan/ ton, and once the excess carbon emissions are calculated by using the modified model, the project cost will increase by 75,000 yuan.

When material recycling is not considered, the carbon emissions at the production stage of building materials reach 10,829.01 tons, which is much higher than the carbon emission value

**Table 6. Carbon emission factors.**

| Materials/energy | Carbon emission factor values | Unit (of measure) |
|---|---|---|
| Diesel fuel | 3.121 | t CO$_{2e}$/t |
| Electrical power | 0.610 | t/MWh |
| Steel reinforcing bar | 2.309 | t CO$_{2e}$/t |
| Low-alloy steel welding rod E43 series | 9.53 | t CO$_{2e}$/t |
| Ready-mixed concrete C20 and C30 | 0.295 | t CO$_{2e}$/m$^3$ |
| Medium and coarse sand | 0.004 | t CO$_{2e}$/t |
| Soil, bentonite | 0.00269 | t CO$_{2e}$/t |
| Clinker | 0.702 | t CO$_{2e}$/t |
| Reinforcing bars up to and including 10 diameters | 2.34 | t CO$_{2e}$/t |
| Paint coatings (general purpose) | 3.5 | t CO$_{2e}$/t |
| High-strength bolt | 2.35 | t CO$_{2e}$/t |
| plate | 2.4 | t CO$_{2e}$/t |
| Steel stranded wire | 2.375 | t CO$_{2e}$/t |
| Ironware | 2.28 | t CO$_{2e}$/t |
| Iron wire | 1.7 | t CO$_{2e}$/t |
| Electrode | 9.53 | t CO$_{2e}$/t |
| Steel shroud | 2.31 | t CO$_{2e}$/t |
| Steel pipe | 2.53 | t CO$_{2e}$/t |
| Caoutchouc | 3.36 | t CO$_{2e}$/t |
| Planking, asphalt sleepers | 0.1463 | t CO$_{2e}$/m$^3$ |
| Diesel | 2.9355 | t CO$_{2e}$/t |
| Labor | 0.00046 | t CO$_{2e}$/workday |

of 4,967.02 tons in the case of nonphysical energy transportation. Although there is high carbon compensation at the production stage of building materials due to the high metal carbon emission factor, the higher the proportion of recyclable materials is, the lower the proportion of carbon emissions at the production stage of building materials to the total carbon emissions.

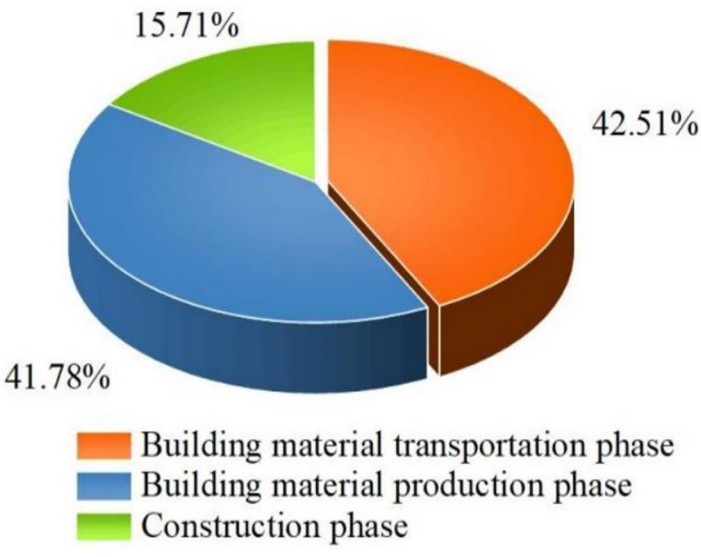

**Fig 1. Carbon emission content share.**

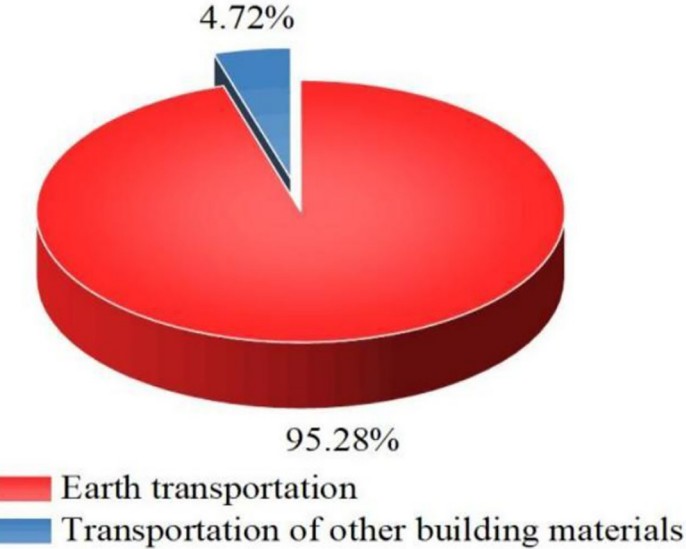

**Fig 2. Carbon emission share at the transportation stage.**

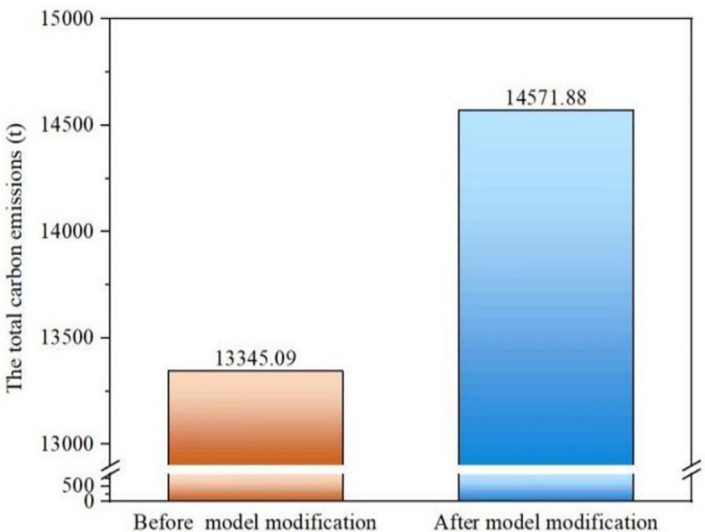

**Fig 3. Comparison of the total carbon emissions before and after model modification.**

In summary, when material recycling and the actual transportation-related energy consumption are not considered, pit project carbon emissions cannot be accurately assessed, which is not conducive to the management of energy conservation and emission reduction in pit project engineering.

## 4.4 Carbon emission reduction measures for excavation engineering

**(1) Design phase.** The foundation pit project is a temporary project, and the previous support design mainly considers the balance between safety and economy. Within the context of the

energy-saving and emission reduction era, support structure selection should consider low-carbon development and environmental protection as the third element, considering safety, economy, and low-carbon aspects.

**(2) Building material production stage.** Focusing on changes in production processes, e.g., Barcelo et al. [31], by improving the production process of cement clinker, the emission factor can be reduced by 30% relative to conventional cement while maintaining the strength.

**(3) Building material transportation phase.** Transportation routes should be reasonably planned to avoid the increase in the fuel consumption of transportation vehicles caused by road factors, and regular maintenance of earth-moving vehicles should be conducted to prevent the wear and tear of the internal parts of vehicles from resulting in increased fuel consumption. In addition, the use of local materials is an important measure to reduce carbon emissions.

**(4) Construction phase.** The necessity of human factor reliability in low-carbon actions should be emphasized. At present, construction site laborers are generally older and exhibit limited low-carbon awareness. The management of construction personnel should be strengthened to promote low-carbon construction processes.

**(5) Dismantling phase.** The recycling of waste building materials should be emphasized. For example, the recycling of metal materials can lead to a life-cycle emission reduction of up to 50% [32].

## 5 Conclusion

In this paper, we calculated carbon emissions originating from excavation engineering under road transportation conditions based on the life cycle assessment and carbon emission process analysis methods, and the following conclusions can be obtained:

1. The carbon emissions of foundation pit projects mainly result from earth transportation. Adopting the 02B foundation pit of the Beijing Urban Subcenter Station Transportation Hub Project as an example, the carbon emissions of earth transportation account for 95.28% of the carbon emissions at the transportation stage, of which earth transportation accounts for 40.50% of the total carbon emissions.

2. The actual fuel consumption of vehicles is considered, resulting in the carbon emissions of excavation engineering increasing by more than 9.2%. Hence, the impact cannot be ignored.

3. Carbon emission models for the transportation phase must account for external factors that affect vehicle fuel consumption, such as congestion levels, temperature conditions, road types, and vehicle loads.

4. The pit project must consider the implicit carbon emissions influenced by the surrounding environment. The carbon emissions resulting from engineering measures such as deformation monitoring of adjacent buildings, reinforcement of the surrounding soil, and protection of underground pipelines should not be ignored.

5. This study uses the Beijing Urban Sub-center Station Transportation Hub project as a case study for carbon emission research. The analysis results can provide a theoretical basis for energy conservation and emission reduction in foundation pit construction. However, considering the heavy workload of material transport vehicles and the internal factors of the vehicles that affect energy consumption, the analysis may still underestimate the actual carbon emissions. In addition, this study analyzes the construction process of a single project. Due to the complexity of the foundation pit construction process, there is uncertainty in

the emission factors of materials used, actual transportation distances, and types of energy consumed, which limits the conclusions drawn. Therefore, future research should consider using uncertainty methods for carbon emission analysis in foundation pit projects to support energy conservation and emission reduction in such projects.

## Supporting information

**S1 File. Beijing construction project pricing basis budget consumption standard (Beijing construction project PBBCS).**
(PDF)

**S2 File. Building carbon emission calculation standard.**
(PDF)

**S3 File. Fuel consumption for trucks in operation.**
(PDF)

**S4 File. National unified machinery shift cost quota.**
(PDF)

## Author Contributions

**Writing – original draft:** Zhiheng Zhang.

**Writing – review & editing:** Bai Chongxi.

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
