## [Decision Letter · Decision Letter 0]

8 Oct 2024

PONE-D-24-19328Carbon emission quantification analysis of pit works under road transport conditionsPLOS ONE

Dear Dr. Chongxi,

Thank you for submitting your manuscript to PLOS ONE. After careful consideration, we feel that it has merit but does not fully meet PLOS ONE’s publication criteria as it currently stands. Therefore, we invite you to submit a revised version of the manuscript that addresses the points raised during the review process.

Based on the opinion of both reviewers, your article needs a major revision. Please make the corrections and suggested items carefully and send the article for review.

We look forward to receiving your revised manuscript.

Kind regards,

Somayeh Soltani-Gerdefaramarzi, Ph. D.

Academic Editor

PLOS ONE

Journal requirements: When submitting your revision, we need you to address these additional requirements. 1. Please ensure that your manuscript meets PLOS ONE's style requirements, including those for file naming. The PLOS ONE style templates can be found at https://journals.plos.org/plosone/s/file?id=wjVg/PLOSOne_formatting_sample_main_body.pdf and https://journals.plos.org/plosone/s/file?id=ba62/PLOSOne_formatting_sample_title_authors_affiliations.pdf 2. Thank you for stating the following financial disclosure:  [This work was supported by the [ National Natural Science Foundation of China ] under Grant [No. 51608101].].  Please state what role the funders took in the study.  If the funders had no role, please state: ""The funders had no role in study design, data collection and analysis, decision to publish, or preparation of the manuscript."" If this statement is not correct you must amend it as needed. Please include this amended Role of Funder statement in your cover letter; we will change the online submission form on your behalf. 3. Please include captions for your Supporting Information files at the end of your manuscript, and update any in-text citations to match accordingly. Please see our Supporting Information guidelines for more information: http://journals.plos.org/plosone/s/supporting-information.  4. Please amend either the title on the online submission form (via Edit Submission) or the title in the manuscript so that they are identical.

Reviewers' comments:

Reviewer's Responses to Questions

**Comments to the Author**

1. Is the manuscript technically sound, and do the data support the conclusions?

Reviewer #1: Partly

Reviewer #2: Yes

2. Has the statistical analysis been performed appropriately and rigorously? 

Reviewer #1: Yes

Reviewer #2: Yes

3. Have the authors made all data underlying the findings in their manuscript fully available?

Reviewer #1: Yes

Reviewer #2: No

4. Is the manuscript presented in an intelligible fashion and written in standard English?

Reviewer #1: Yes

Reviewer #2: No

5. Review Comments to the Author

Reviewer #1: The article deals with an important aspect of the pit construction and its lifecycle. The following issues should be addressed before the acceptance.

The introduction is talking about building construction and its lifecycle which is not the scope of this research. It should be modified accordingly.

Section 2 and 3 can be merged together.

There is no reference for Equations (1), (2), (6) and Tables 1 to 4.

It is not clear what updates the authors made with the emission model? Is it one of the equations mentioned in the previous comment? If yes, then the basis of developing these equations should be mentioned.

The analysis was applied to single project. Hence, the limitations of the work should be mentioned specifically with the conclusions. Accordingly, the future direction and implementation of the present research should be mentioned.

Reviewer #2: Please apply the following suggestions to improve the manuscript:

1. I don't think it is important to include the division of the life cycle of pit construction in the abstract. The abstract should contain general information about the work, the methods used, the results, and the novelty of the study.

2. The calculation model should be included in the keywords.

3. In line 73, the phrase "In a large number of studies at home and abroad" should be revised. I suggest proofreading the entire text to improve the English.

4. It is recommended to emphasize and highlight the novelty of the study in the introduction. While there are short reviews of previous works, more detailed information about the meaning of the life cycle should be provided, as you underline its importance.

5. Please create a "Methods" section where you describe the structure of your analysis and present the workflow. After that, you can describe the remaining aspects.

6. Regarding subsection 3.1: This section could be expanded with a more detailed discussion of recycling and material reuse methodologies. It would be helpful to provide examples of materials that have a significant impact on emissions, such as aluminum compared to concrete.

7. The description of individual variables, such as energy consumption (Pi) and the carbon emission factor (EF), could be further developed. For example, it would be beneficial to explain what kind of energy is being considered (fuel, electricity) and how the emission factor is determined for different fuel types.

8. Regarding the model in subsection 3.3: Unlike other phases analyzed in previous sections, this model does not account for correction factors, such as machine operating conditions, climate factors, or variability in operational efficiency depending on the project. Including these elements could make the model more comprehensive.

9. Please improve the size and quality of the figures.

6. PLOS authors have the option to publish the peer review history of their article (what does this mean?). If published, this will include your full peer review and any attached files.

Reviewer #1: No

Reviewer #2: No

---

## [Author Response · Author response to Decision Letter 0]

8 Nov 2024

The author has responded to all comments in the response to the reviewers

---

## [Decision Letter · Decision Letter 1]

2 Dec 2024

Carbon emission quantification analysis of excavation engineering under road transport conditions

PONE-D-24-19328R1

Dear Dr. Chongxi,

We’re pleased to inform you that your manuscript has been judged scientifically suitable for publication and will be formally accepted for publication once it meets all outstanding technical requirements.

Kind regards,

Somayeh Soltani-Gerdefaramarzi, Ph. D.

Academic Editor

PLOS ONE

Additional Editor Comments (optional):

Reviewers' comments:

Reviewer's Responses to Questions

**Comments to the Author**

1. If the authors have adequately addressed your comments raised in a previous round of review and you feel that this manuscript is now acceptable for publication, you may indicate that here to bypass the “Comments to the Author” section, enter your conflict of interest statement in the “Confidential to Editor” section, and submit your "Accept" recommendation.

Reviewer #1: All comments have been addressed

2. Is the manuscript technically sound, and do the data support the conclusions?

Reviewer #1: Partly

3. Has the statistical analysis been performed appropriately and rigorously? 

Reviewer #1: Yes

4. Have the authors made all data underlying the findings in their manuscript fully available?

Reviewer #1: No

5. Is the manuscript presented in an intelligible fashion and written in standard English?

Reviewer #1: Yes

6. Review Comments to the Author

Reviewer #1: (No Response)

7. PLOS authors have the option to publish the peer review history of their article (what does this mean?). If published, this will include your full peer review and any attached files.

Reviewer #1: No

---

## [Editor Report · Acceptance letter]

13 Dec 2024

PONE-D-24-19328R1 

PLOS ONE

Dear Dr. Chongxi, 

I'm pleased to inform you that your manuscript has been deemed suitable for publication in PLOS ONE. Congratulations! Your manuscript is now being handed over to our production team.

Kind regards, 

on behalf of

Dr. Somayeh Soltani-Gerdefaramarzi 

Academic Editor

PLOS ONE